# OpenReview forum: "RLHS: Mitigating Misalignment in RLHF with Hindsight Simulation"
_ICLR.cc/2025/Conference — Submitted to ICLR 2025_

### Official Review · Reviewer_qi6B · 2024-10-29

**Soundness:** 2
**Presentation:** 3
**Contribution:** 2
**Rating:** 5
**Confidence:** 4

**Summary:**

The paper operates in the problem setting where a human asks a question to a large language model (LLM) and then makes a decision based on the LLM's response. The authors start by noting that when humans are asked to label preferences for LLM outputs, they usually cannot see the downstream consequences of those outputs. This short-term feedback can cause the LLM to produce responses that appear satisfactory but could be misleading or harmful when used for decision-making. To address this, the authors propose reinforcement learning from hindsight simulation (RLHS), where, after an LLM outputs responses for preference labeling, plausible consequences are simulated. When giving feedback, the preference labeler can then observe the original LLM output and its simulated consequences. The key idea is that the value of an LLM's response lies not in the response itself, but in how it influences decision-making. The authors demonstrate that a Marketplace AI chatbot, which recommends products to users, tends to learn deceptive behaviors when fine-tuned using short-term feedback (e.g., reinforcement learning from human feedback, or RLHF). In contrast, this behavior does not emerge when RLHS is applied.

I recommend rejecting this paper due to (1) insufficient novelty in the problem setting and proposed solution, and (2) limited empirical and theoretical results.

**Strengths:**

The authors clearly describe the problem setting, justifying why it is important. The proposed solution, RLHS, intuitively makes sense and is well presented. All figures are visually appealing, and general flow and organization of this paper is strong. The insights that the authors bring attention to are important for the broader RLHF community.

**Weaknesses:**

The problem that this paper addresses is important and potentially under explored, but is not new. The authors assert that their “central insight is that the utility provided by an AI system to a human...is not generally an intrinsic property of the outputs that it generates, but rather a function of their real-world consequences…” This insight and the general problem, however, was explored by Lang et al., who the authors cite. Therefore, I do not see the problem setting or this insight as a novel contribution from this paper. This, of course, is not a weakness in and of itself but is taken as a weakness in combination with the points listed below.

The authors’ proposed solution involves simulating potential outcomes given an LLM’s response, and then presenting those outcomes as well as the LLM’s response to a preference labeler. To the best of my knowledge this approach has not been explored by other academic papers, but feels like an incremental contribution. The authors propose no new algorithmic developments or theoretical findings, but rather, proposes using LLM prompting to provide additional information to preference labelers. In combination with the fact that this problem setting is not new, and weak empirical evidence (detailed below), I am not confident in the significance or originality of the papers contributions.

The authors claim that they “provide substantial empirical evidence that immediate human feedback frequently misrepresents true utility in everyday interaction settings...” Rather, the authors only present empirical evidence that immediate human feedback can misrepresent true utility by designing a specific experimental protocol that induces the described phenomena. I do not see evidence to support the claim that this phenomena happens “frequently” in practice, such as in other tasks used by prior work. To alleviate this weakness, the authors could provide examples from other domains or cite existing literature that demonstrates this phenomenon in different contexts.

Relatedly, the authors only use one domain for evaluating RLHS—introduced by the authors themselves—making it difficult to understand how this approach compares to prior work or how useful it is in practice. I appreciate the human study executed by the authors as an extra dimension of analysis, but there is a lack of sufficient evidence that RLHS is practical to use or beneficial compared to standard RLHF procedures stemming from the narrow domain in which it was evaluated.

The authors also claim that “immediate feedback in online training introduces more misalignment gaps than offline training”, supported by the observation that PPO with immediate feedback yields higher satisfaction ratings but lower true utilities than DPO. The authors do not hypothesize or justify any explanation for this observation. I do not see sufficient information, in the form of either further empirical results, theoretical analysis, or a rigorous explantation, to support this claim that seeks to generalize empirical results for PPO and DPO in one domain to all online and offline training methods. I urge the authors to further investigate a non-intuitive claim like this through experiments in other domains to ensure that it still holds and through ablation studies to better understand it. For example, would we see the same relationship between other offline and online training methods?

In short, I think RLHS is a potentially interesting direction of research, but currently I do not see sufficient novelty in the problem setting or proposed solution, as well as sufficient empirical or theoretical results, to recommend this paper for publication in this venue.

**Questions:**

Questions/Clarifications that could impact the score:
1. Is there a case for why the problem setting of this paper is novel that I am missing?
2. Do the authors have additional evidence or reasoning behind the claim that “immediate feedback in online training introduces more misalignment gaps than offline training”?

Things to improve the paper that did not impact the score:
1. More information on how data was collected via Prolific should be included in the Appendix. This includes but is not limited to the following: How were human subjects chosen? From what population? How were subjects assigned to the experimental conditions?  How were subjects compensated? Was any data removed/filtered out? Did this study receive IRB approval? Without this information, it is difficult for readers to understand the context in which the human study was conducted, and therefore to fully interpret its results.
2. The authors wrote “Hypothesis 1: Models trained with RLHS lead to a higher long-term user satisfaction rate and lower regret rate than those trained with RLHF using immediate feedback”. This is the first time you introduced the terms “long-term user satisfaction rate” and “regret rate”. The former is ambiguous (what constitutes “long-term”), and the latter is undefined (what is “regret rate” and how is this computed?).

---

> ### Author Response · Authors · 2024-11-23
>
> We thank the reviewer for the thoughtful feedback, which provides us with the opportunity to clarify the novelty of our contributions and address the concerns raised.
>
>
> > The problem that this paper addresses is important and potentially under explored, but is not new. The authors assert that their “central insight is that the utility provided by an AI system to a human...is not generally an intrinsic property of the outputs that it generates, but rather a function of their real-world consequences…” This insight and the general problem, however, was explored by Lang et al., who the authors cite. Therefore, I do not see the problem setting or this insight as a novel contribution from this paper.
>
> > Is there a case for why the problem setting of this paper is novel that I am missing?
>
>
>
> Our key insight is that the value of AI outputs lies in their downstream consequences, critically including how they affect the real-world behavior of human users. While the importance of long-term outcomes is a core principle of dynamic decision theory, in the context of RLHF our paper is the first to (1) explore the negative implications of learning from immediate foresight human feedback, and (2) propose a general mitigation strategy by evaluating the real-world downstream harm caused by inaccurate information.
>
> The reviewer refers to a recent paper by Lang et al. which theoretically studied that partial observability of an AI assistant’s task execution during human–AI interaction can lead RLHF to learn deceptive behaviors. However, this is distinct from the problem we study, which focuses on **human misprediction of future outcomes**. In the setting considered by Lang et al., the user’s utility does not depend on the long-term outcomes of the interaction, but takes place in the same time frame as the interaction itself. In their formulation, the AI system is incentivized to conceal information from the human user in order to avoid a negative feedback score, but the analysis **does not consider the real-world impact of such deception on the human’s utility** (the outcome has already happened regardless of whether the human observes it or not). While the paper observes that such deception is undesirable, there is no theoretical analysis of **why** it is undesirable or **what** its resulting damage can be. In contrast, in our setting **the outcome hinges on the decisions made by the human user after interacting with the AI system**, so deceiving the human has direct negative repercussions on their realized utility. Quantifying these repercussions requires us to predict the “closed-loop” evolution of the sociotechnical system formed by the human and the AI, which is not done by Lang et al.
>
> **Our paper is the first to capture this dynamical interplay mathematically in the RLHF formulation, and it is also the first to propose and evaluate a general mitigation method that drastically reduces misalignment and deceptive behavior.** Incidentally, our mitigation method can be effective to address some of the instances of AI deception considered by Lang et al., particularly those in which the human user would be better off if the AI system accurately reported on the outcome of its actions rather than “embellish” it: if the user can make better decisions (e.g. take remedial action) with an accurate diagnosis, RLHS will automatically discourage AI hallucination or deception.
>
> Additionally, we emphasize that our partial hindsight approach still operates within a partially observable setting. The minimal difference between partial and oracle hindsight suggests that the fundamental issue in the class of misalignment we study is not primarily linked to partial observability, but rather to the human misprediction of the *downstream consequences*. We added these additional discussion in **Appendix E.1** in our paper.

---

> ### Author Response · Authors · 2024-11-23
>
> > The authors’ proposed solution involves simulating potential outcomes given an LLM’s response, and then presenting those outcomes as well as the LLM’s response to a preference labeler. This approach has not been explored by other academic papers, but feels like an incremental contribution.
>
> The core insight of the paper is the benefit of hindsight in RLHF. Specifically, we propose that delayed feedback provided after observing the outcome can significantly mitigate misalignment issues. This insight is not incremental but rather addresses a fundamental aspect of RLHF dynamics that has been overlooked in much of the existing literature and engineering practice. Our findings demonstrate that even relatively straightforward modifications can result in substantial improvements, as evidenced by Figures 3, 4, and the results from our human studies. Given the measurable impact of this approach, we believe it is essential to share this contribution with the community.
>
> Additionally, we have expanded our discussion in Appendix E.2 to emphasize the impact of this insight and discuss how to incorporate such insight into real-world applications.
>
> > The authors propose no new algorithmic developments or theoretical findings, but rather, propose using LLM prompting to provide additional information to preference labelers.
>
> Our work encompasses algorithmic development, theoretical innovation, and substantial empirical findings.
>
> **Algorithmic development**: We introduce a novel alignment algorithm called Reinforcement Learning from Hindsight Simulation (RLHS), which serves as an alternative to traditional RLHF methods. By simulating future outcomes before human feedback, RLHS effectively mitigates misalignment, as demonstrated in Figures 3 and 4. Notably, our approach to leveraging hindsight extends beyond simulation, with direct applicability to real-world scenarios, as detailed in Appendix E.2.
>
> **Theoretical innovation**: We provided the theoretical analysis in Section 3 that providing human evaluators with hindsight during RLHF generally reduces misalignment and improves utility. We emphasize that **our problem formulation accounts for the human user’s decisions and actions in the real world** (outside the context of the human–AI interaction), and this is crucial because the utility the human user gets comes from the real world, and not only (or even primarily) from the interaction itself. This is not done in Lang et al. or any prior RLHF work to the best of our knowledge. To further elucidate our insights, we have expanded Section 3 with additional definitions of hindsight and foresight values and their corresponding discussions.
>
> **Empirical findings**: Our simulation results demonstrate the following core insight:
> - There is a significant misalignment using immediate feedback, while partial hindsight mitigates such misalignment.
> - RLHF training with immediate feedback leads to very high satisfaction rate but its true utility is significantly worse than train with RLHS training with hindsight feedback.
>
> Complementing the simulation analysis, we formulate and validate 4 hypotheses in human studies:
> - Models trained with RLHS lead to a higher long-term user satisfaction rate and lower regret rate than those trained with RLHF using immediate feedback.
> - RLHF using immediate feedback often experiences a notable decline in user satisfaction once future outcomes are revealed, while RLHS mitigates this decline.
> - RLHS leads to significantly higher true utility than RLHF
> - Models trained with RLHS are more truthful, presenting a strong correlation between their high immediate user satisfaction rate (subjective) and high true utility (objective)

---

> ### Author Response · Authors · 2024-11-23
>
> > The authors only present empirical evidence that immediate human feedback can misrepresent true utility by designing a specific experimental protocol that induces the described phenomena. I do not see evidence to support the claim that this phenomena happens “frequently” in practice, such as in other tasks used by prior work. To alleviate this weakness, the authors could provide examples from other domains or cite existing literature that demonstrates this phenomenon in different contexts.
>
> Our experimental protocol is designed to allow us to readily quantify the ground-truth user reward obtained from the purchase. However, the observed behavior is not induced by our experimental protocol, but can be observed in today’s LLM behavior across various platforms and services. For example: Amazon’s Rufus AI helper chatbot frequently makes mistakes by hallucinating positive responses to questions about unknown product features [1, 2]. Furthermore, prior work has extensively studied the sycophantic behavior of LLMs [3, 4, 5, 6], where models are optimized to generate responses that align with user beliefs rather than the truth. Sharma et al. [6] showed that both humans and preference models (PMs) often prefer convincingly written sycophantic responses over accurate ones a significant portion of the time.
>
> To address these concerns, we have expanded our discussion in **Appendix E.1** to include relevant work. In **Appendix E.2**, we explain why these shortcomings exist in real-world applications and how our proposed approach can address that.
>
> [1] Kaziukėnas, J. (2024b, November 6). Amazon’s Shopping AI Is Confident and Wrong. Marketplace Pulse. https://www.marketplacepulse.com/articles/amazons-shopping-ai-is-confidently-wrong
>
> [2] Dieter. “Amazon’s AI Shopping Assistant Rufus Is Often Wrong.” ConsumerAffairs, 7 Nov. 2024, www.consumeraffairs.com/news/amazons-ai-shopping-assistant-rufus-is-often-wrong-110724.html. Accessed 20 Nov. 2024.
>
> [3] Cotra, Ajeya. "Why AI alignment could be hard with modern deep learning." Cold Takes (2021).
>
> [4] Perez, Ethan, et al. "Discovering language model behaviors with model-written evaluations." arXiv preprint arXiv:2212.09251 (2022).
>
> [5] Wei, Jerry, et al. "Simple synthetic data reduces sycophancy in large language models." arXiv preprint arXiv:2308.03958 (2023).
>
> [6] Sharma, Mrinank, et al. "Towards understanding sycophancy in language models." arXiv preprint arXiv:2310.13548 (2023).
>
> > Do the authors have additional evidence or reasoning behind the claim that “immediate feedback in online training introduces more misalignment gaps than offline training”?
>
> We thank the reviewer for the questions and suggestions. To provide additional empirical support for our claim, we conducted further hypothesis significance tests and evaluated an additional offline RLHF approach (SimPO), comparing it with the online RLHF method (PPO).  These new results and analyses have been included in **Appendix A**, along with minor adjustments to our previous analysis and claims.
>
> We ran both t-tests and a two-way ANOVA to better understand emergent misalignment and the effectiveness of mitigation through hindsight simulation under online and offline fine-tuning schemes. Comparing online and offline fine-tuning methods, we found that PPO with immediate feedback yields significantly lower true utility for the user than DPO ($p  = 1.1 \times 10^{-4}$ in t-test). In addition, considering the difference between the (normalized) user rating and true utility, we find that immediate feedback in online RLHF using PPO introduces a larger misalignment gap than offline RLHF using DPO ($p  = 6.7 \times 10^{-5}$ in t-test). Incorporating partial hindsight helps mitigate this misalignment gap across online and offline fine-tuning ($p = 3.1 \times 10^{-116}$ in two-way ANOVA test).
>
> Additionally, we compared online PPO and offline SimPO. Our analysis revealed that PPO introduces a larger misalignment gap than SimPO ($p = 8.2 \times 10^{-5}$ in t-test ), while partial hindsight effectively reduces misalignment in SimPO as well ($p = 5 \times 10^{-56}$ in t-test).
>
> | Metric             | PPO (IF)       | PPO (PHS)      | SimPO (IF)     | SimPO (PHS)    |
> |--------------------|--------------|--------------|--------------|--------------|
> | **Rating ↑**       | 0.97 ± 0.021 | 0.41 ± 0.026 | 0.94 ± 0.032 | 0.37 ± 0.028 |
> | **True Utility ↑** | -0.71 ± 0.029 | 0.18 ± 0.025 | -0.49 ± 0.044 | 0.16 ± 0.032 |

---

> ### Author Response · Authors · 2024-11-23
>
> >  More information on how data was collected via Prolific should be included in the Appendix. This includes but is not limited to the following: How were human subjects chosen? From what population? How were subjects assigned to the experimental conditions? How were subjects compensated? Was any data removed/filtered out? Did this study receive IRB approval? Without this information, it is difficult for readers to understand the context in which the human study was conducted, and therefore to fully interpret its results.
>
> Thank you for the suggestion. We have added the following in **Appendix D.2**, and add a sentence to the main text in Section 6 so that readers know to refer to it.
>
> The human subjects were chosen from a high quality Prolific participant pool, where participants were pre-screened to have an approval rate of 95-100 over at least 100 previous submissions. Participants were located in the USA.
>
> To assign subjects to experimental conditions, we used random assignment, and each participant was only assigned to one shopping scenario (either one purchasing decision or comparing between two AI shopping assistants). As a negative experience could bias participants’ perceptions of AI chatbots, we ensured that they were not able to retake the study.
>
> The expected duration of the study was 5 minutes, and actually completed the study at a median time of 4:54. Subjects were compensated \\$1.10 for their participation, resulting in a hourly wage of \\$13.47/hour, which was substantially higher than minimum wage.
>
> In addition to participant satisfaction ratings or preferences, participants were asked to provide a brief 2-sentence explanation to explain their ratings or preferences. We manually reviewed these explanations for all participants, and participants that did not provide a reasonable 2-sentence explanation had their data removed from the study. We also removed participants that finished the study in an unreasonably short time (<1:30 out of the estimated 5 minutes). Other than this, no data was removed.
>
> This study received IRB approval at the institution that we conducted the study at. However, due to double-blind rules we cannot provide more information. We are happy to add this to the paper after it is accepted.
>
> > The authors wrote “Hypothesis 1: Models trained with RLHS lead to a higher long-term user satisfaction rate and lower regret rate than those trained with RLHF using immediate feedback”. This is the first time you introduced the terms “long-term user satisfaction rate” and “regret rate”. The former is ambiguous (what constitutes “long-term”), and the latter is undefined (what is “regret rate” and how is this computed?).
>
> We thank the reviewer for the question. To clarify, the hindsight rating measures user satisfaction after considering hindsight information, which reflects long-term satisfaction. Additionally, in our human study interface, participants who purchased an item are given the option to either keep or return it after receiving the product. This step allows us to quantify the regret rate, which represents the proportion of participants who choose to return the item due to dissatisfaction. We have updated our paper to define these terms more clearly.

---

> ### Comment · Reviewer_qi6B · 2024-11-27
>
> Thanks for all of your responses. I am keeping my score as is. I summarize my reasoning below:
>
> The authors state they are the first to
> > (1) explore the negative implications of learning from immediate foresight human feedback, and (2) propose a general mitigation strategy by evaluating the real-world downstream harm caused by inaccurate information.
>
> With regards to (1) this feels like a very small, and predictable, contribution on top of Lang et al. In fact, I would still argue that the problem of partial-observability studied by Lang et al., encompasses the problem you study of "human misprediction of future outcomes." Therefore, I still do not see this problem setting as novel to this paper.
>
> With regards to (2) the mitigation strategy feels impractical and non-rigorous; it simply involves giving the preference labeler more information about the task and possible outcomes. Presumably, if an RLHF practitioner observed the undesired affects that the authors characterize, and had the data available to provide annotators with extra information to mitigate this problem, they would do that already.
>
> > Our paper is the first to capture this dynamical interplay mathematically in the RLHF formulation
>
> Other reviewers have pointed this out, but Theorem 1 feels trivial and already known by the RLHF community. Prior papers in the classic preference based RL setting have studied the effects of trajectory-segment length on learning from human preferences.
>
> I appreciate you incorporating some of my feedback, including additional statistical results, but my concerns above are at the core of the score I gave.

---

> > ### Author Response · Authors · 2024-11-28
> >
> > > With regards to (1) this feels like a very small, and predictable, contribution on top of Lang et al. In fact, I would still argue that the problem of partial-observability studied by Lang et al., encompasses the problem you study of "human misprediction of future outcomes." Therefore, I still do not see this problem setting as novel to this paper.
> >
> > We thank the reviewer for their response. We respectfully disagree with the claim that “the problem of partial observability studied by Lang et al. encompasses the problem of human misprediction of future outcomes”. We insist that our problem setting is fundamentally different from Lang et al., to the extent that **it is impossible to derive our misalignment model or its mitigation method from their theoretical analysis.** We elaborate on the fundamental differences between our two formulations to make it clear how neither one contains the other.
> >
> > **A. Human observations.** Quoting Lang et al., their analysis assumes “that the observations only depend on the environment state, and not directly on the agent’s actions”. Conversely, in our formulation the human’s observations during the interaction come directly from the AI outputs (e.g., the human reads the text generated by the LLM).
> >
> > **B. AI information advantage.** In Lang et al., the AI system is assumed to have full state information (the AI policies are a function of the true state), whereas the human has only partial observations of the state. In our own work, both the human and the AI system have partial state information and, in fact, the AI system may have equal or even less information than the human: our analysis still allows for RLHF-incentivized deception in such cases (and indeed we observe it empirically in our results), by the AI hallucinating information that the human seeks.
> >
> > **C. Timeframe.** The analysis in Lang et al. considers a self-contained assistance problem where the human’s utility is fully realized between the initial and final moment of the interaction. In particular, this utility is not affected by later events that may take place after the interaction. Conversely, our problem formulation hinges on downstream consequences: inaccurate beliefs are harmful to the extent that they favor later human decisions that result in poor outcomes; if the utility was not affected by post-interaction events, the AI could do no harm by deceiving the human.
> >
> > **D. Role of the human.** Lang et al. model the human as a passive observer and not as an agent making decisions in an environment beyond the AI system interface. In contrast, we model the human as an agent facing a partially observable *decision process*.
> >
> > **E. Role of the AI system.** The analysis in Lang et al. requires the AI to take actions that intrinsically produce utility for the human during the timeframe of the interaction (such as installing a software package). Without this assumption, the human would have nothing to evaluate in the Lang et al. analysis, since human feedback is assumed to score the tasks that the AI has carried out. In contrast, we do not assume that the AI takes any actions impacting the world state or explicitly generating utility: we are precisely concerned with communicative actions (which Lang et al. intentionally exclude from the analysis; see point A above) and focus on their implicit value in terms of preparing the human to obtain downstream returns after the interaction is over.
> >
> > As a result of these differences, Lang et al. are only able to show theoretically that the AI system has an incentive to deceive the human in their setting, but they are not able to quantify whether or to what extent this deception negatively impacts the human’s own utility, and they instead appeal to the common-sense notion that deception is generally undesirable. No results in Lang et al. suggest that a solution to this deceptive behavior could come from simulating the follow-up human decisions and their resulting returns during fine-tuning, because such downstream consequences are outside their problem formulation. In our work, these downstream human decisions are crucial: they allow us to quantify the misalignment in terms of human loss of real-world utility and hypothesize a “Goodhart’s law dynamic” in RLHF whereby the human’s true downstream utility will gradually go down as their immediate satisfaction goes up during AI fine-tuning—a hypothesis that we then validate computationally and through human user studies. Our mitigation strategy is based on the simulation of these AI-induced human decisions and their downstream consequences at RLHF training time, so it is neither small nor predictable from Lang et al., because that work *explicitly* excludes (i) AI-dependent human observations, (ii) human actions, and (iii) post-interaction consequences.

---

> > > ### Author Response · Authors · 2024-11-28
> > >
> > > Beyond the above key differences in problem formulation (time frame, agent roles, and information structure), we emphasize that our novelty with respect to Lang et al. also lies in our methodology and scope: While Lang et al. present purely theoretical results (only applied to two handcrafted toy MDP examples and no foundation models), our work performs extensive empirical evaluations with mainstream LLMs, including both simulation experiments and human studies. These experiments demonstrate the practical applicability and effectiveness of our proposed framework, providing a comprehensive analysis beyond what Lang et al. explored. In particular, we provide (i) empirical evidence of RLHF systematically driving an increase in deceptive behavior—and an accompanying decrease in true utility—when used to fine-tune mainstream LLMs (Llama 2-7b, Llama 3-8b); (ii) a concrete, computationally inexpensive mitigation strategy; and (iii) validation of the identified phenomenon and our proposed mitigation through user studies with real human participants. None of this is minor: as shown in our extended human user study results, Llama 3-8b fine-tuned with standard RLHF in the marketplace setting hallucinated in 80% of user queries, while RLHS fine-tuning completely removed hallucinations (0% of queries) while achieving major improvements in user utility (Fig. 5) and closing 85–90% of RLHF’s dramatic alignment gap visible in Figs. 3 and 4.

---

> > > > ### Author Response · Authors · 2024-11-28
> > > >
> > > > > With regards to (2) the mitigation strategy feels impractical and non-rigorous; it simply involves giving the preference labeler more information about the task and possible outcomes. Presumably, if an RLHF practitioner observed the undesired affects that the authors characterize, and had the data available to provide annotators with extra information to mitigate this problem, they would do that already.
> > > >
> > > > The reviewer appears to be under the impression that our proposed method is providing the evaluator with additional data about true consequences that we are somehow making available—that is not what we are doing. Indeed, RLHF practitioners have no practical way to access real data about “ground truth” states or the downstream outcomes of AI outputs. The crucial feature of our method is that it does not require any privileged information about *real* states or outcomes—it only relies on a general-purpose “world model”, which (i) may be the same or a different generative AI model and (ii) need not itself be aligned, to *sample* possible states and *simulate* plausible consequences. That is, we are not “cheating” by introducing any “real” data that wouldn’t be available to evaluators.
> > > >
> > > > While at first glance it may seem like randomly sampling possible outcomes prior to obtaining evaluator feedback would not be particularly helpful, it actually plays a critical role by *decoupling* outcome prediction from the (possibly inaccurate) beliefs induced by the AI output. In conventional RLHF, evaluators’ assessment of the value provided by an AI output to a user typically requires predicting its helpfulness to the user’s later endeavors (what we call “foresight feedback”), but this prediction may be *contaminated* by the AI output misleading the evaluators to believe the prospects are better than they truly are. In other words, evaluators’ predictions of user actions and their effects can both be influenced to some degree by the AI output. Instead, in our hindsight simulation, the predicted human **actions** are conditioned on the AI-induced belief, but their **effects** are not. Thanks to this, whenever the AI system lies or even hallucinates unknown information (recall that, unlike Lang et al., we typically do not grant the AI full state information), our hindsight simulation will, in expectation, yield a poor outcome (since it will predict the human acting on this information “at face value”, but not always with the desired results). As long as the evaluator can accurately assess outcomes when presented with them directly (what we call “hindsight feedback”), then we have an unbiased estimator of the desired training signal.
> > > >
> > > > **We emphasize that feedback providers do not typically have access to the truth when providing feedback.** These points are discussed in detail in Appendix E.2, and we encourage the reviewer to revisit these arguments for further clarity.

---

> > > > > ### Author Response · Authors · 2024-11-28
> > > > >
> > > > > > Other reviewers have pointed this out, but Theorem 1 feels trivial and already known by the RLHF community. Prior papers in the classic preference based RL setting have studied the effects of trajectory-segment length on learning from human preferences.
> > > > >
> > > > > While Theorem 1 certainly follows the familiar structure of other RLHF works, its significance is actually novel (as highlighted by reviewer [bevH]). It specifically looks at the medium and long-term consequences of the human–AI interaction and, in cases where the difference in downstream rewards between two candidate interactions dies off over time (i.e., the consequences do not ripple far into the user’s future), provides a surprisingly strong result for RLHS: Provided an accurate “world model”, conditioning hindsight feedback on **random, simulated consequences** at training time allows completely circumventing the foresight-induced misalignment and recovers an arbitrarily accurate estimate of the reward function, akin to that of standard RLHF in the (unrealistic) no-consequences setting.
> > > > >
> > > > > Importantly, whenever post-interaction consequences matter (there are significant reward differences for different human behaviors at times $t>T$), the same result is not true of standard RLHF (with $N=0$): its estimate of the human reward will generally be biased and no amount of feedback elicitation will resolve the issue. In other words, RLHS is the right formulation of RLHF in settings where the human user and evaluator have imperfect information about the world.

---

### Official Review · Reviewer_HpbZ · 2024-10-31

**Soundness:** 3
**Presentation:** 4
**Contribution:** 3
**Rating:** 8
**Confidence:** 3

**Summary:**

This paper addresses misalignment issues in generative AI models trained with Reinforcement Learning from Human Feedback (RLHF), which typically relies on immediate feedback that may not account for long-term effects on user satisfaction. The authors show that this shortsighted feedback can lead to issues like sycophancy and deception in model behavior. To address this, they introduce Reinforcement Learning from Hindsight Simulation (RLHS), an approach that simulates potential downstream consequences and gathers feedback based on those outcomes. Applied to Proximal Policy Optimization (PPO) and Direct Preference Optimization (DPO), RLHS significantly reduces misalignment. A user study demonstrates that RLHS leads to greater goal achievement and satisfaction, highlighting the value of focusing on simulated long-term outcomes in alignment strategies.

**Strengths:**

Originality: Original idea that addresses a real problem with RLHF (which is well described)

Quality: Experiments very well designed with both LLMs simulating humans and real humans, and with both PPO and DPO, to support the claims of the paper.

Clarity: Very well written and understandable, from the problem formulation, to the explanation of the method.

Significance: Could have significant impacts in specific settings where this kind of hindsight simulation data is available

**Weaknesses:**

The quality of the results in this paper seems to rely heavily on the quality of the simulation for hindsight feedback. I think there should be an analysis of using different sizes of models for the hindsight feedback and see what kind of effect this has on performance.

The evaluation is also only conducted in one environment (a shopping environment) where the RLHS algorithm makes a lot of sense. I think there should be at least one other environment used in the paper to demonstrate the generality of the method to different domains.

**Questions:**

Do the authors have any results on the effect of the quality of the simulation on performance?

Could the authors demonstrate another environment where this algorithm might be useful?

---

> ### Author Response · Authors · 2024-11-23
>
> > The quality of the results in this paper seems to rely heavily on the quality of the simulation for hindsight feedback. I think there should be an analysis of using different sizes of models for the hindsight feedback and see what kind of effect this has on performance.
>
>
> We thank the reviewer for this insightful feedback. The hypothesis regarding the quality of LLM feedback and its alignment with human feedback is valid and important. To investigate this, we conducted an additional human study to assess how closely the feedback and actions of Llama-3.1-70B align with those of human participants. In the study, participants interacted with chatbots from two different stores, taking actions such as purchasing items or leaving the store based on the conversations. After engaging with both stores, participants were asked to choose which store they preferred. We randomly sampled 10 scenarios from our training set, with 30 different participants evaluating each scenario. To determine the human preference for each scenario, we employed majority voting. This method was used to ensure that the aggregated choice reflected the consensus among participants, minimizing the impact of individual variability and providing a more robust measure of overall preference. Our analysis revealed that the matching accuracy between LLM-generated feedback and human feedback reached $100\\%$.Furthermore, the actions taken by the LLM matched those of human participants with $95\\%$ accuracy. These findings suggest that Llama-3.1-70b is powerful enough to provide simulated feedback that aligns closely with the human. We provided additional information of this study in Appendix D.
>
> > The evaluation is also only conducted in one environment (a shopping environment) where the RLHS algorithm makes a lot of sense. I think there should be at least one other environment used in the paper to demonstrate the generality of the method to different domains.
>
> We thank the reviewer for this thoughtful comment. We agree that evaluating RLHS across multiple environments would further demonstrate the generality of our method. However, due to resource and time constraints, we focused on the shopping environment as a representative domain where user decision-making and feedback play a critical role. This environment was selected because it allows us to quantify the ground-truth user reward obtained from purchases, thereby providing a clear and quantitative demonstration of the benefits of hindsight feedback in mitigating misalignment. We believe that the core principles of RLHS—leveraging hindsight feedback to improve alignment—can be adapted and applied to other environments, and we have included a discussion on the broader applicability of RLHS in other settings in the Broader Impact section (Appendix E.2).

---

> > ### Comment · Reviewer_HpbZ · 2024-11-24
> >
> > Thank you for your clarifications! I maintain that I think this is a good paper and keep my score unchanged

---

> > > ### Author Response · Authors · 2024-11-25
> > >
> > > Thank you for taking the time to review our paper. We appreciate your positive feedback and are glad that our clarifications were helpful!

---

### Official Review · Reviewer_Pa2D · 2024-11-02

**Soundness:** 1
**Presentation:** 2
**Contribution:** 2
**Rating:** 5
**Confidence:** 4

**Summary:**

The paper studies the problem of RL finetuning with hindsight, where the model collects feedback based on the ultimate outcome rather than intermediate preference.

**Strengths:**

The paper discusses an interesting perspective using the example of marketplace chatbot, where the authors argue that the preference with responses should be collected after the final outcome instead of directly based on chat experience. This can potentially be a way to help fix the data labeling issues. The authors also provide some experimental results showing that misalignment is mitigated here.

**Weaknesses:**

I have a few major concerns regarding the paper.

1. I'm not fully convinced by the motivation and solution. It seems from the example of marketplace that for the case of deceptive response, the model actually does not follow the instructions and provided responses that are in conflict with the given context. Compared with asking for end outcomes which comes with high uncertainty, it seems that one can easily ask human labeler (or a strong LLM proxy) to label if the responses are consistent with the given context or not. It's also unclear how one can track the end outcome given that on marketplace, one may browse different items and then take completely different actions than purchase. To me it appears to be more of a data labeling issue that can be fixed with some easier way.

2. How is the theory related to any aspects of the practice? The theory appears to be a trivial fact that because of the discount factor used for estimating utility, it suffices to estimate utility in a short time period of time. Are we sure this is consistent with the case in practice, where the authors argue that the end outcome is most important? What can we learn from the theorem here? It appears to be here so that there will be some theoretical analysis in the paper.

3. The main issue the paper wants to resolve seems to be that the model is not following given context but instead hallucinates some non-existing objects. This seems to be also easily fixable by tuning on some good instruction following / preference dataset in public, which will force the model to follow given context better. Indeed, I would be surprised if this still happens in modern LLMs like llama 3.1 70B, GPT-4o, Claude 3.5 etc., as they have been already trained to follow instructions qutie well. It's also unclear to me why people want to only tune the chatbot on the satisfcation rate data, instead of mixing with other good instruction following dataset which shall resolve the problem easily.

**Questions:**

Please see weakness section above.

---

> ### Author Response · Authors · 2024-11-23
>
> We thank the reviewer for the thoughtful feedback, which provides us with the opportunity to clarify the novelty of our contributions and address the concerns raised.
>
> > Compared with asking for end outcomes which comes with high uncertainty, it seems that one can easily ask human labeler (or a strong LLM proxy) to label if the responses are consistent with the given context or not. To me it appears to be more of a data labeling issue that can be fixed with some easier way.
>
> The main point we want to emphasize is that human evaluators do not always know the full truth when providing feedback. This limitation poses significant challenges for real-world applications of AI, particularly within the RLHF framework we studied.  We discussed these limitations in Appendix E.2 and how our proposed hindsight feedback approach can help overcome them to enhance AI alignment.
>
> - **Limited Access in Real-World Applications**: In real-world scenarios, users and human labelers often interact with black-box or closed-prompt AI systems, where the internal prompts and decision-making processes are not fully transparent. Notable Examples include commercial systems like OpenAI's ChatGPT and Amazon's Rufus. The techniques we propose (hindsight feedback), and our experimental setting can be directly applied to these systems when full internal access is unavailable. In such cases, assessing consistency alone is insufficient as external context might not capture the complete implications of an AI's output.
>
> - **Limitations of Human Judgment and Information Access**: Even when human evaluators have full access to models and their prompts (e.g., in open-source systems), perfect judgment is not guaranteed. Imperfect evaluators might miss deeper implications or fail to predict the long-term impact of a response. Below, we outline two practical examples illustrating these limitations and how hindsight feedback can address them:
>
>     **Code Generation Scenario**: Imagine a user asking a language model for code to fit a polynomial curve to a set of data points. One solution may fit the data perfectly, while another shows some deviation. A human evaluator might prefer the model with the perfect fit, not realizing that it overfits and performs poorly on new data. Immediate feedback in this case could lead to misalignment, as it prioritizes surface-level satisfaction over long-term utility. By providing feedback after testing the code on new data (hindsight), evaluators can offer more informed input, reducing misalignment. Hindsight simulation can automate part of this process by allowing models to test outcomes on unseen data and report the results to human evaluators for better feedback. *One extra benefit of hindsight simulation is that humans do not need to be domain experts to provide truthful feedback.*
>
>     **AI4Science Proof Construction**: When constructing mathematical proofs for scientific problems, misaligned models often generate results that are correct only under conditions or assumptions specified by the user. Human evaluators, constrained by time or limited expertise, may overlook these limitations during evaluating the model, eventually causing the model to overfit to a restricted set of problems and unable to tackle scientific problems in general settings. On the other hand, hindsight simulation generates a diverse set of scenarios, including, e.g., edge cases, under which the model is required to validate its proof. This allows the human evaluator to assess the model performance based on its ability to generalize beyond the immediate problem. This approach ensures that the constructed proof concept is not only mathematically valid for problems in the dataset, but also robust and broadly applicable.
>
> In our main paper, we used a marketplace chatbot to *quantitatively* demonstrate the misalignment phenomenon and to provide empirical evidence supporting the effectiveness of our mitigation strategies. However, the broader goal of our work is to introduce the benefit of hindsight feedback and suggest its integration into current RLHF pipelines. As illustrated in the examples above, hindsight feedback has the potential to address critical limitations in human evaluation processes across diverse applications.

---

> ### Author Response · Authors · 2024-11-23
>
> > How is the theory related to any aspects of the practice? The theory appears to be a trivial fact that because of the discount factor used for estimating utility, it suffices to estimate utility in a short time period of time. Are we sure this is consistent with the case in practice, where the authors argue that the end outcome is most important? What can we learn from the theorem here? It appears to be here so that there will be some theoretical analysis in the paper.
>
> Thank you for your feedback. We would like to clarify how our theory is closely connected to practical applications.
>
> Our theory highlights the benefit of hindsight, demonstrating that: (1) the difference in human feedback between one given by a Boltzmann-rational human evaluator and one with the ideal infinite-horizon oracle decreases monotonically as the hindsight step ($N$) increases, and (2) this difference becomes arbitrarily small as $N$ approaches infinity.
>
> This theoretical insight is directly tied to practical outcomes, as demonstrated in our marketplace chatbot system. In our experiments, immediate feedback (hindsight horizon $N=0$) resulted in significant misalignment, as indicated by the huge gap between human rating and utility in the left of Fig.3 and 4. However, incorporating partial hindsight ($N=1$) substantially mitigated this issue (middle of Fig.3 and 4). Notably, the performance of partial hindsight closely matched that of the oracle hindsight (right of Fig.3 and 4). This suggests that adding even a using additional hindsight step ($N=1$) is effective at estimating true human utility and reducing misalignment. Our empirical results validate the theory, and we strongly recommend to practitioners working on foundation models to incorporate at least one hindsight step in RLHF to mitigate  misalignment issues.
>
> To further elucidate our insights, we have expanded Section 3 with additional definitions on hindsight and foresight values and their corresponding discussions. We discussed the fundamental reason for misalignment due to human feedback based on their less grounded foresight value (Definition 3 in the updated paper). Hindsight simulations, on the other hand, allow human evaluators to assess outcomes based on their long-term, experienced outcome rather than immediate observations and perceived future outcomes. This allows them to provide feedback based on the more grounded hindsight value (Definition 2 in the updated paper), leading to a better-aligned reward model. We hope these additions address the reviewer's concerns.

---

> ### Author Response · Authors · 2024-11-23
>
> > The main issue the paper wants to resolve seems to be that the model is not following given context but instead hallucinates some non-existing objects. This seems to be also easily fixable by tuning on some good instruction following / preference dataset in public, which will force the model to follow given context better. Indeed, I would be surprised if this still happens in modern LLMs like llama 3.1 70B, GPT-4o, Claude 3.5 etc., as they have been already trained to follow instructions qutie well. It's also unclear to me why people want to only tune the chatbot on the satisfcation rate data, instead of mixing with other good instruction following dataset which shall resolve the problem easily.
>
> We thank for your feedback and would like to clarify the focus of our paper.
>
> - The main issue we aim to address is not hallucination or the model’s ability to follow context accurately but rather the misalignment issue in RLHF. We provided empirical evidence that immediate feedback frequently misrepresents the human’s true utility in consultancy-type interactions and, when used as a proxy for it in RLHF fine-tuning, it systematically drives misalignment with human goals. To address this, we propose to leverage *hindsight* as a simple but effective misalignment mitigation mechanism, in which evaluators experience the downstream outcomes of an interaction before being asked for feedback on the model.
> - Fine-tuning on high-quality instruction-following or preference datasets could help reduce hallucination, but this is distinct from the topics and insights explored in our paper. Importantly, *the fundamental problem of RLHF misalignment we studied is not related to, nor can it be resolved through, fine-tuning on additional datasets*. Instead, it stems from the limitations of human evaluators, who often lack access to the full truth when providing feedback. We have added a detailed discussion in Section E.2 to explain why this problem is particularly challenging in real-world applications and how our proposed insight can help mitigate it.
> - Our marketplace experimental setting serves as an illustrative example that allows us to quantify user outcomes objectively, such as rewards obtained from purchases. The key contribution of our work is showing that delaying feedback until after outcomes are known can mitigate the misalignment. In our experiments, we used both Llama-2-7b and Llama-3-8B, which were already fine-tuned on a large amount of public data. Notably, while the initial true utility of Llama-3-8B was high, fine-tuning with immediate feedback resulted in a significant decrease in utility and an increase in the misalignment gap. Conversely, fine-tuning with hindsight feedback mitigated this misalignment, demonstrating the effectiveness of our proposed approach. While we were not able to fine-tune a large model like GPT-4, we propose that our corel insight should be applied to any language model fine-tuned with RLHF to enhance its alignment with human values and long-term satisfaction.

---

> > ### Comment · Reviewer_Pa2D · 2024-11-26
> >
> > Thank you for providing the comments. I'm on the same page that the misalignment phenomenon exists in the real world. From my point of view, such hindsight feedback is a different way of feedback data collection that tries to mitigate the error or bias in the original immediate human preference data. I appreciate the authors' efforts in identifying the problems, and I have changed my score accordingly to reflect this part of contribution.
> >
> > However, I'm not fully convinced how the current solution could generalize in a meaningful and nontrivial way to the scenarios the authors discussed, like AI4Sci proof and code generation. First, the proposed solution in the paper for the market place example is just to use a stronger LLM to simulate decision making process and relabel the data with CoT. It doesn't seem to be generalizable to other cases. Second, RLHF based on deterministic reward signal like code execution results have become quite standard in the field, which does not appear to be novel to me. For the example of AI4Sci proof, it seems that the best proposed solution is just to let a stronger model to do more CoT for covering the edge cases. All of these seem to be mostly fixing data labeling error with some specifically designed tricks, rather than through some general principles.

---

> ### Author Response · Authors · 2024-11-28
> **Response to additional questions**
>
> > However, I'm not fully convinced how the current solution could generalize in a meaningful and nontrivial way to the scenarios the authors discussed, like AI4Sci proof and code generation. First, the proposed solution in the paper for the market place example is just to use a stronger LLM to simulate decision making process and relabel the data with CoT. It doesn't seem to be generalizable to other cases. Second, RLHF based on deterministic reward signal like code execution results have become quite standard in the field, which does not appear to be novel to me. For the example of AI4Sci proof, it seems that the best proposed solution is just to let a stronger model to do more CoT for covering the edge cases. All of these seem to be mostly fixing data labeling error with some specifically designed tricks, rather than through some general principles.
>
> We thank the reviewer for the thoughtful discussion. We are offering some further clarifications below and have further updated the manuscript to include a new table and algorithm.
>
> We agree that applying additional techniques such as chain of thought (CoT) can improve the performance of the model and reduce its likelihood of producing hallucinations. However, we would like to emphasize that the human-independent problem of logically sound AI outputs is distinct from the human-bound problem of AI alignment. While advancements in these two directions are certainly complementary, neither is sufficient by itself. We recall that RLHS particularly targets the misalignment induced by malleable human foresight in RLHF. We use the marketplace setting as our main example because it offers a clear context in which evaluators may be misled by AI outputs (as can users). We welcome the opportunity to further discuss the code generation and AI4Sci proof settings, more subtle tasks where the value of RLHS is perhaps less evident but, we believe, still significant.
>
> As we emphasize in the paper, users often seek help from AI chatbots precisely in areas where they lack expertise, and our examples involving code generation and AI4Science proof construction aim to illustrate this point. A novice math/programming student might run with the (technically correct) proof/code generated by an LLM in the context of a specific query, only to later struggle to use it in the full context of their class project. The student’s immediate feedback would fail to reflect the fact that the generated proof/code turned out hard to adapt for its true intended application. RLHS may discourage AI models from producing correct but poorly reusable outputs by systematically simulating—at training time—plausible future problems the student, based on the context provided, may need to tackle with the generated code/proof, and predicting the student’s ability to extend it to meet these new needs that they failed to foresee in their original query. An aligned AI system would interpret the novice student’s request pragmatically and, rather than give them exactly what they asked for, might ask some follow-up questions and suggest that the student go with a more future-proof version, or accompany the output with some helpful tips on how to extend it. To clarify the general structure of the RLHS scheme, we have added an algorithm box in Appendix E.2 and included Table 7 with examples of how the algorithm can be applied in different settings.

---

### Official Review · Reviewer_bevH · 2024-11-04

**Soundness:** 2
**Presentation:** 3
**Contribution:** 4
**Rating:** 5
**Confidence:** 4

**Summary:**

The authors note that RLHF relies on immediate feedback, which often fails to capture long-horizon patterns well, causing behaviors such as sycophancy and deception. RLHS mitigates these issues by doing "hindsight simulation", in which they are simulating plausible future outcomes and providing feedback on those plausible outcomes. The effectiveness of RLHS is empirically validated by showing significant reductions in misalignment, improvements in user satisfaction & true utility in real marketplace chatbots.

**Strengths:**

- The paper extends our understanding of RLHF by showing that hindsight horizon $N$ exponentially improves the accuracy of utility estimates. Lemma 1 & Theorem 1 show that the difference in finite-hindsight utility estimation approaches the true utility difference as $N$ increases. This is a unique and strong theoretical result.
- Applicable to both PPO and DPO-like approaches.
- True utility and satisfaction rating are well-defined as concepts to evaluate.
- Figures 1-5 and Tables 1-2 are visually well-presented, easy to read and clear.
- Evidence to believe that this approach might even extend to other approaches and domains that use preference learning.
- Leads to consistent improvements in true utility even under partial hindsight (Figure 3 & Table 1).
- Includes actual human study, which is mostly well-performed.

**Weaknesses:**

- Assumes human evaluators operate under $P(s \succ s') = \sigma(\beta (R_T(s) - R_T(s')))$, and completely discounts the possibility of error correction, bounded rationality and cognitive biases. Should at least propose how hindsight simulations would deal with systematic bias/errors and discuss those effects in higher detail.
- Should discuss mathematically and empirically the sensitivity of RLHS to different values of the hindsight horizon $N$.
- The evaluation utility $(U = 0$, $U = -1$, $U = 1$) is too simplistic, and the work should consider e.g. systematic risk aversion terms and other concepts from bounded rationality to deal with e.g. time to reach Boltzmann-like decisions and other more realistic considerations.
- Justification for normalization of satisfaction ratings must be clearer.

**Questions:**

- Why normalize satisfaction ratings between -1 and 1, as opposed to e.g. using the Likert scale? Please also consider adding an analysis of the distributions of ratings under different training conditions.
- Do you have empirical results on how RLHS is sensitive to different $N$? E.g. do you have any empirical work to determine optimal values for N?

---

> ### Author Response · Authors · 2024-11-23
>
> We thank the reviewer for the insightful feedback and for acknowledging the uniqueness of our theoretical results and the significance of our empirical contributions.
>
> > Assumes human evaluators are Boltzmann rational decision makers, and completely discounts the possibility of error correction, bounded rationality and cognitive biases. Should at least propose how hindsight simulations would deal with systematic bias/errors and discuss those effects in higher detail.
>
> Thank you for the insightful comment! Our work focuses specifically on mitigating misalignment in the reward caused by humans providing feedback based on their less grounded foresight value (Definition 3 in the updated paper). Hindsight simulations allow human evaluators to assess outcomes based on their long-term impact rather than immediate observations, and provide feedback based on the hindsight value (Definition 2 in the updated paper), which is much more grounded and would lead to a better-aligned reward model. We would like to respectfully point out that the issue of misalignment caused by human feedback based on foresight is fundamentally different from misalignment caused by human bounded rationality or cognitive biases, which pertain to the inherent limitations or systematic errors in human decision-making. We anticipate that foresight could be intrinsically connected to bounded rationality or cognitive biases. However, studying such synergy is beyond the scope of this paper. We believe hindsight simulation could complement future research efforts in understanding misalignment caused by bounded rationality or cognitive biases, and we look forward to exploring these synergies in subsequent research.
>
> > Should discuss mathematically and empirically the sensitivity of RLHS to different values of the hindsight horizon
>
> > Do you have empirical results on how RLHS is sensitive to different N? E.g. do you have any empirical work to determine optimal values for N?
>
> In our experimental setup, partial hindsight corresponds to the case where N=1. We show empirically that even with a single step of hindsight, the misalignment is significantly reduced, demonstrating the effectiveness of our approach.
>
> Oracle hindsight approximates the scenario where N is sufficiently large, allowing the user to observe the outcome for all relevant products. While the performance of oracle hindsight is better than partial hindsight, the difference is not substantial in the online marketplace environment.
>
> We would like to point out that the sensitivity of RLHS to the hindsight horizon is closely tied to the reward curve of the specific problem (figure 13 in the appendix E)—tasks with reward peak appearing later in the future may benefit from a longer hindsight horizon, while others suffice with shorter ones (e.g., the marketplace example in our paper). Therefore, we strongly recommend that practitioners working on foundation models incorporate at least one hindsight step in RLHF to mitigate misalignment issues.
>
> > The evaluation utility $(U = 0, U = -1, U = 1)$ is too simplistic, and the work should consider e.g. systematic risk aversion terms and other concepts from bounded rationality to deal with e.g. time to reach Boltzmann-like decisions and other more realistic considerations.
>
> Thank you for the insightful comment. Our evaluation utility is chosen in a way that it can demonstrate the core benefit of RLHS in a simple, yet effective way. While we agree that adding the suggested risk aversion terms and accounting for human-bounded rationality could potentially improve the validation in real-world settings, we focused on this simple, easy-to-grasp utility definition to highlight the impact of hindsight simulation on improved outcomes in the marketplace setting.

---

> ### Author Response · Authors · 2024-11-23
>
> > Why normalize satisfaction ratings between -1 and 1, as opposed to e.g. using the Likert scale?
>
> We thank the reviewer for the question, which allows us to clarify our motivation. We chose to normalize satisfaction ratings to a scale between -1 and 1 to align them with the scale of true utility, which also ranges from -1 to 1. This alignment facilitates direct comparison and clearer visualization of the misalignment phenomenon within a single plot. We have incorporated these new explanations in the paper.
> We also recognize the importance of presenting results using the original Likert scale for additional clarity. To address this, we have included these results in **Appendix A (Figure 8, 9, 10)**.
>
> > Please also consider adding an analysis of the distributions of ratings under different training conditions.
>
> We thank the reviewer for this valuable suggestion. In response, we have included a histogram analysis of the ratings for Llama-2-7b trained with both immediate feedback and hindsight simulation in **Appendix A**.  We observed that models trained with immediate feedback achieve very high satisfaction ratings (predominantly 5). However, this comes at the expense of true utility (-0.71), which remains negative and underscores the misalignment issue between perceived satisfaction and actual performance. Training with hindsight feedback still maintains a high satisfaction rating while significantly improving true utility, achieving positive values (+0.18). This indicates that partial hindsight mitigates the misalignment, resulting in more truthful model performance.

---

> > ### Author Response · Authors · 2024-12-01
> >
> > We thank the reviewer for the insightful feedback and for acknowledging the uniqueness of our theoretical results and the significance of our empirical contributions.
> >
> > We enhanced our empirical results by including Likert scale satisfaction ratings for Llama-2-7b and Llama-3-8b, along with histograms of the ratings, and corresponding analysis. Additionally, we have added further discussions about our theoretical results.
> >
> > As the rebuttal period is coming to a close, we would greatly appreciate it if the reviewer could kindly address our responses to your original review.

---

### Author Response · Authors · 2024-11-23

We would like to thank all the reviewers for their helpful comments and suggestions. We are delighted that all reviewers acknowledged the importance of our insight. We also appreciate the reviewer for highlighting our theoretical contribution (“unique and strong theoretical result”), and strong empirical results in both simulation and human study (“Experiments very well designed with both LLMs simulating humans and real humans”).

We appreciate this opportunity to address the reviewers’ questions and provide corresponding updates to the manuscript (shown in blue in the revised version). Specifically, we made the following changes:
- We further clarified the novelty of our contribution in Appendix E.1 and discussed related work in greater detail. In particular, we added more explanations to the problem we studied and discussed the fundamental difference between our problem formulation and theoretical insights with that of Lang et al. We also made adjustments in the introduction to better clarify our key insights.
- We expanded the discussion of real-world applications of our insights in Appendix E.2. We emphasized the challenges of obtaining truthful feedback in real-world settings and how our proposed method effectively addresses them.
- We clarified our key theoretical insights on hindsight simulation mitigates misalignment with additional mathematical definitions of foresight and hindsight values in Section 3.1 and added the corresponding discussions.
- We enhanced our empirical results by including Likert scale satisfaction ratings for Llama-2-7b and Llama-3-8b, along with histograms of the ratings. We provided additional statistical significance results for comparing online and offline fine-tuning and added new results on SimPO to strengthen our findings in Appendix A.
- We conducted an additional human study to validate that the feedback and actions of our AI proxy closely align with those of human participants, as detailed in Appendix D.3.
- We provided additional details on our human study in Appendices D.1 and D.2, metrics in Sections 4.2 and 6, and analysis in Section 6 and Appendix A.

Please note that throughout our responses we are using the numbers of, e.g., equations and definitions, corresponding to those in the updated manuscript.

We look forward to continued discussion with the reviewers.

---

### Meta-Review · Area_Chair_E7a2 · 2024-12-29

**Metareview:**

This paper introduces RL with Hindsight Simulation (RLHS), a method to mitigate misalignment in RLHF by simulating plausible future outcomes and providing feedback on those. The authors tested the proposed method upon both PPO and DPO in several examples.


The major issues raised by most of the reviewers lies in following aspects:

1, Novelty: The long-term accumulated reward is extensively exploited in classic RL, and the core insight in exploiting utility depends on real-world consequences is similar to prior work (Lang et al.).

2, Weak Empirical Evaluation: The major empiricaly comparison is conducted on a special domain that is designed by the authors on purpose, therefore, the prevalence of the motivation in real-world setting is not clear, and the signficance of the proposed method is difficult to evaluate.

3, There are several unjustified claims (“immediate feedback in online training introduces more misalignment gaps than offline training”) as pointed out by the reviewers.

In sum, the paper can be further improved by taking the reviewers' suggestion into account.

**Additional Comments On Reviewer Discussion:**

The authors provides additional experiments. However, these experiments still focus on the problem designed by the authors, therefore, it does not address the major issues raised by the reviewers.

---

### Decision · Program_Chairs · 2025-01-22

Reject